# Short-Lived Antibody-Mediated Saliva Immunity against SARS-CoV-2 after Vaccination

Johannes Roth Madsen,[a] Bettina Eide Holm,[a] Laura Pérez-Alós,[a] Rafael Bayarri-Olmos,[a] Anne Rosbjerg,[a] Kamille Fogh,[b,c] Mia Marie Pries-Heje,[d] Dina Leth Møller,[e] Cecilie Bo Hansen,[a] Line Dam Heftdal,[e,f] Rasmus Bo Hasselbalch,[b,c] Sebastian Rask Hamm,[e] Ruth Frikke-Schmidt,[g,h] Linda Hilsted,[h] Susanne Dam Nielsen,[e,g] Kasper Karmark Iversen,[b,c,g] Henning Bundgaard,[d,g] Peter Garred[a,g]

[a]Laboratory of Molecular Medicine, Department of Clinical Immunology, Section 7631, Rigshospitalet, University of Copenhagen, Copenhagen, Denmark

[b]Department of Cardiology, Herlev and Gentofte Hospital, University of Copenhagen, Copenhagen, Denmark

[c]Department of Emergency Medicine, Herlev and Gentofte Hospital, University of Copenhagen, Copenhagen, Denmark

[d]The Heart Center, Department of Cardiology, Rigshospitalet, University of Copenhagen, Copenhagen, Denmark

[e]Viro-immunology Research Unit, Department of Infectious Diseases, Section 8632, Rigshospitalet, University of Copenhagen, Copenhagen, Denmark

[f]Department of Haematology, Rigshospitalet, University of Copenhagen, Copenhagen, Denmark

[g]Department of Clinical Medicine, Faculty of Health and Medical Sciences, University of Copenhagen, Copenhagen, Denmark

[h]Department of Clinical Biochemistry, Rigshospitalet, University of Copenhagen, Copenhagen, Denmark

**ABSTRACT** Knowledge about the effect of vaccination against severe acute respiratory syndrome coronavirus 2 (SARS-CoV-2) on immunity reflected in the saliva is sparse. We examined the antibody response in saliva compared to that in serum 2 and 6 months after the first vaccination with the BNT162b2 vaccine. Four hundred fifty-nine health care professionals were included in a prospective observational study measuring antibody levels in saliva and corresponding serum samples at 2 and 6 months after BNT162b2 vaccination. Vaccinated, previously SARS-CoV-2-infected individuals (hybrid immunity) had higher IgG levels in saliva at 2 months than vaccinated, infection-naive individuals ($P < 0.001$). After 6 months, saliva IgG levels declined in both groups ($P < 0.001$), with no difference between groups ($P = 0.37$). Furthermore, serum IgG levels declined from 2 to 6 months in both groups ($P < 0.001$). IgG antibodies in saliva and serum correlated in individuals with hybrid immunity at 2 and 6 months ($\rho = 0.58$, $P = 0.001$, and $\rho = 0.53$, $P = 0.052$, respectively). In vaccinated, infection-naive individuals, a correlation was observed at 2 months ($\rho = 0.42$, $P < 0.001$) but not after 6 months ($\rho = 0.14$, $P = 0.055$). IgA and IgM antibodies were hardly detectable in saliva at any time point, regardless of previous infection. In serum, IgA was detected at 2 months in previously infected individuals. BNT162b2 vaccination induced a detectable IgG anti-SARS-CoV-2 RBD response in saliva at both 2 and 6 months after vaccination, being more prominent in previously infected than infection-naive individuals. However, a significant decrease in salivary IgG was observed after 6 months, suggesting a rapid decline in antibody-mediated saliva immunity against SARS-CoV-2, after both infection and systemic vaccination.

**IMPORTANCE** Knowledge about the persistence of salivary immunity after SARS-CoV-2 vaccination is limited, and information on this topic could prove important for vaccine strategy and development. We hypothesized that salivary immunity would wane rapidly after vaccination. We measured anti-SARS-CoV-2 IgG, IgA, and IgM concentrations in saliva and serum in both previously infected and infection-naive individuals, 2 and 6 months after first vaccination with BNT162b2, in 459 hospital employees from Copenhagen University Hospital. We observed that IgG was the primary salivary antibody 2 months after vaccination in both previously infected and infection-naive individuals, but dropped significantly after 6 months. Neither IgA nor IgM was detectable in saliva at either time point. Findings indicate that salivary immunity against SARS-CoV-2 rapidly declines following vaccination in both previously infected and infection-naive

Address correspondence to Peter Garred, peter.garred@regionh.dk.

The authors declare no conflict of interest.

individuals. We believe this study shines a light on the workings of salivary immunity after SARS-CoV-2 infection, which could prove relevant for vaccine development.

**KEYWORDS** SARS-CoV-2 vaccination, saliva, IgG, IgA, IgM, immunity

During infection, the first barrier that severe acute respiratory syndrome coronavirus 2 (SARS-CoV-2) encounters is the mucosal surface in the airways (1). Here, SARS-CoV-2 uses its spike protein to infect the host's cells. The spike protein binds to the ACE-2 receptor on cells in the respiratory tract and is primed by the cell's TMPRSS2 protease, which results in the fusion of the viral and cellular membrane (2).

The body has different ways of protecting itself against pathogens, one of which is neutralizing antibodies on the mucosal surface. Neutralizing antibodies can be secreted on the mucosal surface and bind to pathogens to inhibit their entry into the host's cell (3). The dominant antibody on mucosal surfaces is IgA, produced by plasma cells near the mucosal surface and transported through the mucosal epithelium to the lumen via the polymeric immunoglobulin receptor (pIgR) (3, 4). Even though IgA is the dominant antibody on mucosal surfaces, not all mucosal surfaces contain the pIgR. Therefore, they are unable to transport IgA to the surface. The surfaces that do not contain the pIgR, such as the oral cavity, have the neonatal Fc receptor (FcRn), capable of transporting IgG from serum to the lumen (3). Nevertheless, the protection against pathogens invading the mucosal surface is mediated not only by antibodies but also through cellular immunity, like tissue-resident T cells (5).

To protect against SARS-CoV-2 infection, vaccines—for the most part targeting the spike protein—were developed and used worldwide. So far, they have proven to be safe and highly efficacious in preventing severe disease and hospitalization (6). All currently available vaccines are administered intramuscularly. Thus, there is no direct immunological stimulation of the mucosa. The vaccines generally induce a protective response after two doses, and infected individuals achieve similar responses after only one dose (7–9). With the Delta and the Omicron variants of SARS-CoV-2, the vaccines still protect against severe disease, but they are ineffective against onward transmission (10–13). Breakthrough infections have been observed even after three and four doses and in individuals with detectable antibody levels (14–16). If the vaccines induce antibodies at the site of entry of SARS-CoV-2, this could potentially neutralize the virus before it infects the host and, consequently, limit the spread of the disease.

This dichotomy raises the question of how efficient the vaccines are at providing an efficient response on saliva-coated surfaces. Therefore, we investigated which specific anti-SARS-CoV-2 antibodies (IgG, IgA, and IgM) were present in saliva and the corresponding serum samples after vaccination with the BNT162b2 COVID-19 mRNA vaccine (Pfizer-BioNTech) in a prospective cohort of health care personnel at 2 and 6 months after first vaccination dose. Moreover, saliva has been suggested to some degree to be an accessible biomarker of mucosal immunity (17). To elucidate this question, we developed a sensitive Luminex-based anti-spike receptor-binding domain (RBD) antibody assay to investigate saliva. Serum samples were measured using an anti-RBD antibody enzyme-linked immunosorbent assay (ELISA).

## RESULTS

**Luminex assay development (validation cohort).** To validate and determine cut-offs in the IgG and IgM saliva assays, Luminex assays were performed on samples from 16 coronavirus disease 2019 (COVID-19)-convalescent and 24 infection-naive individuals. Thirteen COVID-19-convalescent and 23 infection-naive individuals were included to validate the IgA levels. All samples in the validation cohort were collected before vaccination.

The convalescent group had a mean IgG level of 2.57 (standard error of the mean [SEM], 0.146) $\log_{10}$(fluorescence intensity [FI]), a mean IgA level of 2.57 (SEM, 0.125) $\log_{10}$(FI), and a mean IgM level of 1.73 (SEM, 0.133) $\log_{10}$(FI). The infection-naive group had a mean IgG level of 1.39 (SEM, 0.152) $\log_{10}$(FI), a mean IgA level of 2.15 (SEM,

**TABLE 1** Demographic data and characteristics from the vaccinated study cohort[a]

| Characteristic | Total | | | 2 mo | | 6 mo | |
|---|---|---|---|---|---|---|---|
| | Total (n = 459) | Infection naive (n = 427) | Previously infected (n = 32) | Infection naive (n = 412) | Previously infected (n = 29) | Infection naive (n = 189) | Previously infected (n = 14) |
| Sex | | | | | | | |
| Female | 403 (88) | 376 (88) | 27 (84) | 364 (88) | 25 (86) | 169 (89) | 12 (86) |
| Male | 56 (12) | 51 (12) | 5 (16) | 48 (12) | 4 (14) | 20 (11) | 2 (14) |
| | | | | | | | |
| Age (yrs) | 48 (39–58) | 49 (40–58) | 42 (33–52) | 49 (40–58) | 43 (32–52) | 49 (40–57) | 35 (32–43) |
| BMI[b] | 24 (22–27) | 24 (22–27) | 24 (20–25) | 24 (22–27) | 23 (20–24) | 24 (21–26) | 24 (22–24) |
| | | | | | | | |
| Days from vaccination | | | | | | | |
| First | 63 (59–66) | 63 (59–66) | 63 (61–67) | 62 (59–65) | 63 (60–66) | 170 (160–180) | 160 (150–170) |
| Second | 32 (30–37) | 32 (30–37) | 34 (31–39) | 32 (29–36) | 33 (31–38) | 140 (130–140) | 130 (120–140) |

[a]Values for the participants' sex are number (percent) of individuals; all other are median (IQR).
[b]BMI, body mass index. A total of 61 values are missing.

0.078) $\log_{10}$(FI), and a mean IgM level of 1.64 (SEM, 0.097) $\log_{10}$(FI). The convalescent group had significantly higher IgG and IgA levels than the uninfected unvaccinated group ($P < 0.0001$ and $P = 0.026$, respectively) (see Fig. S1 in the supplemental material). There was no significant difference between the IgM levels in the convalescent group and the infection-naive group (Fig. S1).

Receiver operating characteristic (ROC) curves were utilized to calculate the optimal cutoffs between convalescent and infection-naive individuals in the Luminex-based assays. The cutoff for IgG was determined to be 2.12 $\log_{10}$(FI) with a sensitivity and specificity of 87.5% and 95.8%, respectively. IgA had a cutoff of 2.87 $\log_{10}$(FI) with a sensitivity and specificity of 30.7% and 91.3%, respectively. IgM had a cutoff of 2.25 with a sensitivity and specificity of 12.5% and 91.6%, respectively.

**Overview of included participants in the prospective vaccine study (vaccination cohort).** A total of 644 samples from 459 participants were included in the study. Of them, 570 samples (88.5%) were from 403 female participants (87.8%). The median age among the participants was 48 (IQR, 39 to 58) years for both time points, being somewhat lower in the previously infected group (42 [IQR, 33 to 52] years; $P = 0.027$). The median times between the first vaccination and the 2-month and 6-month samples were 62 (IQR, 59 to 65) days and 170 (IQR, 160 to 180) days, respectively. Of the 459 participants, 274 (59.7%) contributed one sample and 185 (40.3%) contributed two samples. In 601 (93.3%) of the samples, antibodies against nucleocapsid (N) protein were not detected (Table 1).

**IgG levels in saliva and serum.** In saliva, the mean level of anti-RBD IgG in participants previously infected with SARS-CoV-2 was 3.47 (SEM, 0.093) $\log_{10}$(FI) versus 2.97 (SEM, 0.028) $\log_{10}$(FI) in the infection-naive participants ($P < 0.0001$) 2 months after vaccination. The mean level of IgG in saliva in the previously infected participants was 2.58 (SEM, 0.180) $\log_{10}$(FI) versus 2.40 (SEM, 0.064) $\log_{10}$(FI) in the infection-naive participants ($P = 0.37$) 6 months after vaccination. Both the previously infected and the infection-naive participants had a significant decrease in mean IgG level between the 2- and 6-month samples ($P = 0.0003$ and $P < 0.0001$, respectively) (Fig. 1a). In saliva, 38 (88.37%) of the samples in the previously infected group were above the cutoff (2.12 $\log_{10}$[FI]), and 521 (86.69%) of the samples from the infection-naive group were above the cutoff.

In serum, the previously infected group had a serum anti-RBD IgG level of 4.72 (SEM, 0.067) $\log_{10}$(arbitrary units [AU]/mL) versus 4.32 (SEM, 0.019) $\log_{10}$(AU/mL) in the infection-naive group ($P < 0.0001$) 2 months after vaccination. The mean antibody level was 4.03 (SEM, 0.147) $\log_{10}$(AU/mL) in the previously infected group versus 3.41 (SEM, 0.027) $\log_{10}$(AU/mL) in the infection-naive group ($P = 0.00095$) 6 months after vaccination. Serum IgG antibody levels in both participants with hybrid immunity and infection-naive participants decreased significantly between 2 and 6 months postvaccination ($P = 0.0004$ and $P < 0.0001$, respectively) (Fig. 1b). For serum, all samples from

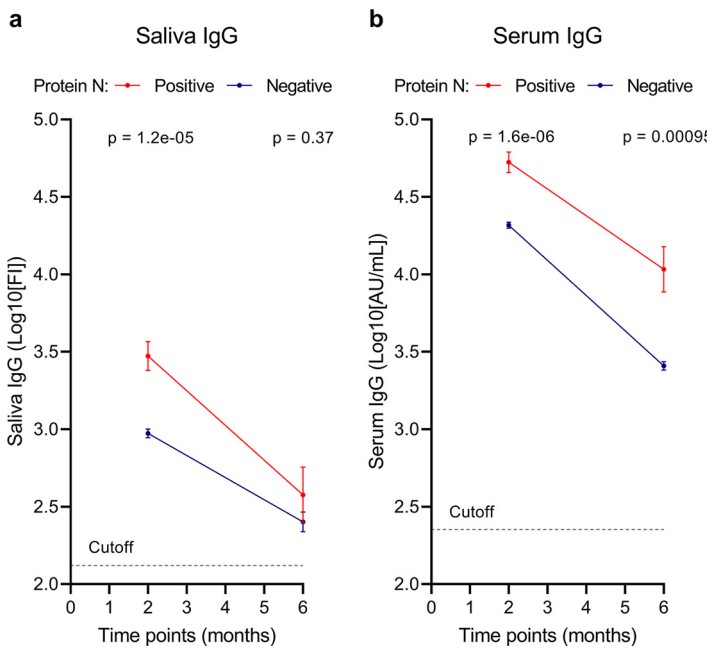

**FIG 1** Evolution of IgG levels in saliva and serum 2 and 6 months after the first vaccine dose. IgG levels in saliva (a) and serum (b) 2 and 6 months after the first vaccine dose. The dashed line shows the cutoff for assay positivity. The blue line represents individuals with natural SARS-CoV-2 infection based on antibodies against N protein. The red line represents SARS-CoV-2 infection-naive individuals based on the absence of antibodies against N protein. Error bars show SEM. *P* values represent differences between infected and uninfected individuals. *P* values of <0.05 are considered significant.

participants with hybrid immunity (*n* = 43) were seropositive and 599 (99.67%) of the samples from infection-naive individuals were seropositive.

There was a significant positive correlation between saliva and serum IgG 2 months after vaccination ($\rho$ = 0.45, *P* < 0.0001) (Fig. 2a). The correlation was weaker 6 months after the vaccination ($\rho$ = 0.16, *P* = 0.027) (Fig. 2b). When the correlation between saliva and serum IgG in the previously infected and the infection-naive groups was investigated, a significant correlation was found in both groups 2 months after the vaccination ($\rho$ = 0.58, *P* = 0.0012, and $\rho$ = 0.42, *P* < 0.0001, respectively) (Fig. 2c and e, respectively). After 6 months, almost the same correlation was observed in the previously infected group ($\rho$ = 0.53, *P* = 0.052), and no correlation was observed in the infection-naive group ($\rho$ = 0.14, *P* = 0.055) (Fig. 2d and f, respectively).

**IgA and IgM levels in saliva and serum.** Interestingly, the mean IgA levels in saliva were below the positivity threshold in the previously infected group as well as in the infection-naive group at 2 and 6 months after the first vaccination dose, with no significant differences between groups (*P* = 0.17 and *P* = 0.22 for 2 and 6 months after vaccination, respectively) (Fig. 3a). Only 4 samples in the previously infected group (9.30%) and 79 samples in the infection-naive group (12.98%) were above the cutoff. Therefore, the results regarding the presence of saliva IgA should be interpreted carefully, as the IgA levels were rather low and close to the assay cutoff.

In serum, levels of IgA antibodies were higher in the previously infected group than the infection-naive group at 2 and 6 months after vaccination (Fig. 3b). Serum anti-RBD IgA levels at 2 months were 2.82 (SEM, 0.128) $\log_{10}$(AU/mL) and 1.74 (SEM, 0.0516) $\log_{10}$(AU/mL) in the previously infected and infection-naive groups, respectively (*P* < 0.0001) (Fig. 3b). At 6 months after vaccination, the serum IgA levels were 1.51 (SEM, 0.368) $\log_{10}$(AU/mL) and 0.35 (SEM, 0.056) $\log_{10}$(AU/mL) in the previously infected and infection-naive groups, respectively (*P* = 0.0080) (Fig. 3b). Both groups had a significant decrease in the level of serum IgA antibodies between the two time points (*P* = 0.0039 and *P* < 0.0001 in previously infected and infection-naive groups,

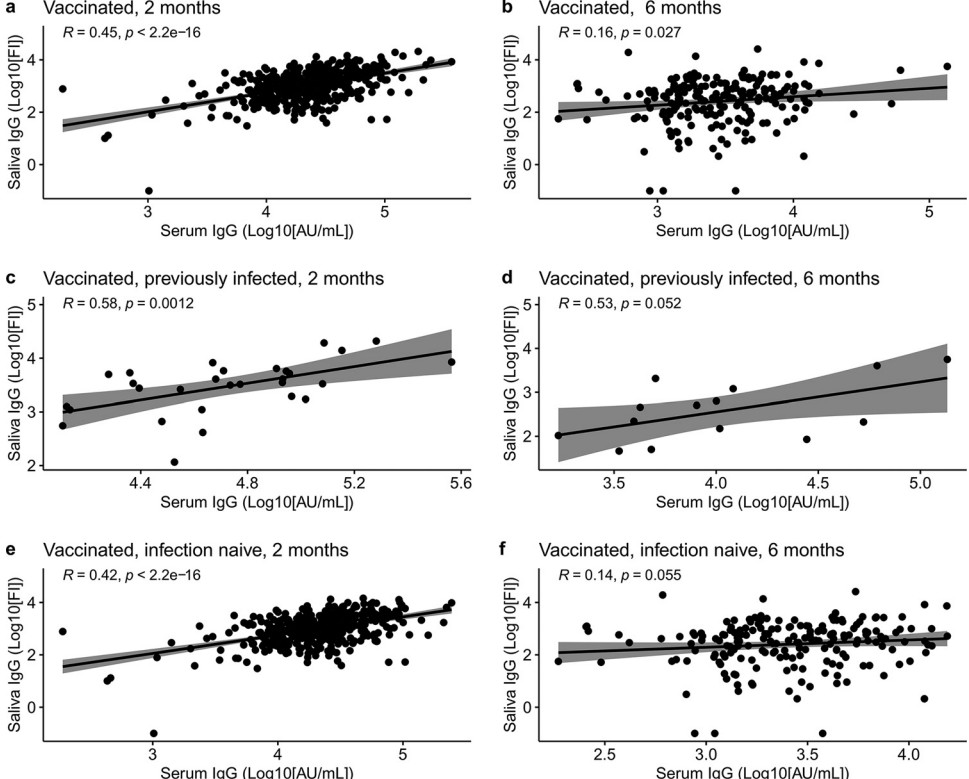

**FIG 2** Correlation of IgG antibodies in serum and saliva with a Spearman rank test. (a and b) Total vaccinated population correlation of IgG antibodies in serum and saliva at 2 and 6 months after the first vaccine dose, respectively. (c and d) Correlation of IgG in serum and saliva in previously infected individuals at 2 and 6 months after the first vaccine dose, respectively. (e and f) Correlation of IgG in serum and saliva in infection-naive individuals at 2 and 6 months after the first vaccine dose, respectively. $P$ values of <0.05 are considered significant.

respectively). For serum, 36 (85.7%) of the samples from the previously infected group were above the cutoff (2 $\log_{10}$[AU/mL]) and 349 (58.07%) of the samples from the infection-naive group were above the cutoff.

Anti-RBD IgM levels were found to be below the cutoff at 2 and 6 months after the first vaccination in both groups and in both fluids (saliva and serum) (Fig. 4). Although IgM levels were similar and below the positivity threshold in saliva (Fig. 4a), significantly higher IgM levels, albeit still below the positivity threshold, were observed in serum in previously infected individuals than in infection-naive individuals at 2 and 6 months after first vaccination ($P = 0.0056$ and $P = 0.0039$, respectively) (Fig. 4b).

## DISCUSSION

The currently approved vaccines are administered intramuscularly, and the humoral response to the BNT162b2 vaccine has been extensively studied (7, 8, 18–20). The vaccine generates a robust IgG and T cell response, decreasing the likelihood of severe disease and fatal outcomes (6). However, breakthrough infections often happen even after the third booster dose (15). This could be due partly to a lack of optimal mucosal immunity. Thus, whether antibody levels in the blood can be used to assess the immune response in mucosal compartments, such as saliva, is not fully elucidated.

To address this question, we developed a sensitive bead-based immunoassay to measure saliva anti-RBD IgG, IgA, and IgM levels. To validate the assay, we compared saliva from a group of unvaccinated COVID-19-convalescent individuals with that from unvaccinated infection-naive individuals (validation cohort). The IgG levels were significantly elevated in the convalescent group, while none in the infection-naive group had IgG levels above the threshold. The difference in IgA levels between the convalescent group and the infection-naive group was significant, but to a lower degree than for

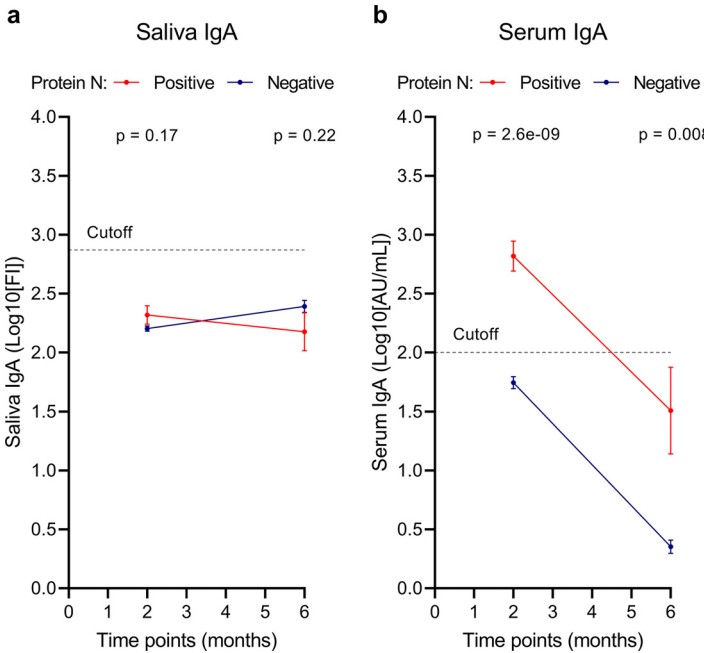

**FIG 3** Evolution of IgA levels in saliva and serum 2 and 6 months after the first vaccine dose. IgA levels in saliva (a) and serum (b) 2 and 6 months after the first vaccine dose. The dashed line shows the cutoff for assay positivity. The blue line represents participants with natural SARS-CoV-2 infection based on antibodies against N protein. The red line represents SARS-CoV-2 infection-naive individuals based on the absence of antibodies against N protein. Error bars show SEM. *P* values represent differences between infected and uninfected individuals. *P* values of <0.05 are considered significant.

IgG, and most samples in both groups were below the assay threshold. No significant difference between the groups was seen for IgM. The results clearly showed that IgG and IgA saliva antibodies to SARS-CoV-2 infection could be measured in the validation cohort, with IgG being more prominent than IgA. We attribute this to different factors: (i) the collected saliva is derived primarily from areas in the mouth cavity covered with squamous stratified epithelial cells, which mainly secrete IgG, and not columnar epithelial cells secreting IgA and IgM, making IgG more prevalent (3), and (ii) IgA appears to have a shorter half-life (21). Therefore, sampling needs to be performed shortly after infection to detect IgA (21).

In our vaccination cohort, we observed that both previously infected and infection-naive individuals generated an IgG response in matched saliva and serum samples. Our study shows that previously infected individuals have a higher level of IgG antibodies in both saliva and serum 2 months after the first vaccination than infection-naive individuals. This difference between groups was observed only in serum and not in saliva 6 months after the first vaccination. The finding that previously infected individuals had a higher level of IgG antibodies is consistent with our observation in the convalescent-phase samples from unvaccinated COVID-19 patients used for validation of the assay and with other studies (22, 23). Furthermore, IgG in saliva and in serum correlated in the previously infected individuals at 2 and 6 months after the first vaccination, whereas IgG levels in the infection-naive individuals correlated only at 2 months after the first vaccination, when the IgG peak is observed after the administration of the second vaccine dose (24). Our data show that the salivary IgG level is highest among the previously infected individuals but decreases rapidly in both groups 6 months after the first vaccination. This indicates that IgG might be a vital isotype in the local oral immune defense after vaccination. We did not observe IgA levels above the assay threshold in saliva in the previously infected or the infection-naive group after vaccination. This contrasts with the results in our assay validation among unvaccinated COVID-19-convalescent individuals. The reason for this discrepancy is unclear, but it

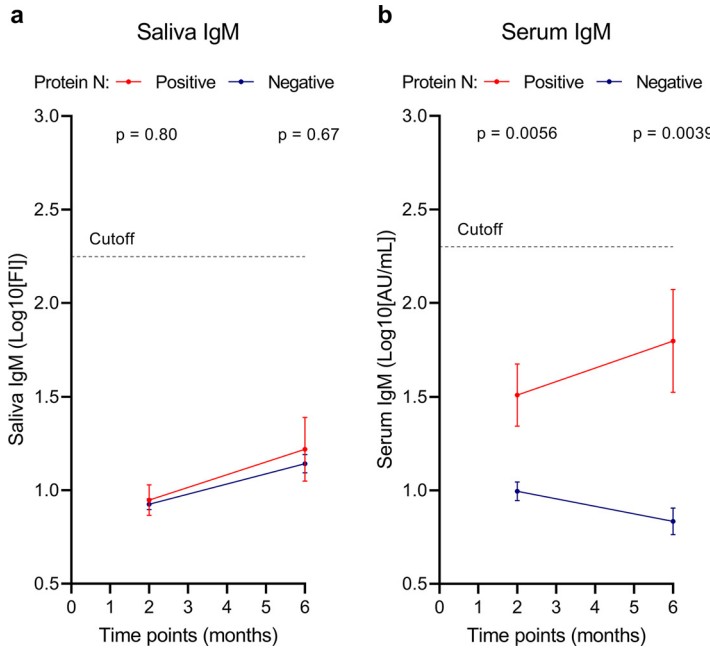

**FIG 4** Evolution of IgM levels in saliva and serum 2 and 6 months after the first vaccine dose. IgM levels in saliva (a) and serum (b) 2 and 6 months after the first vaccine dose. The dashed line shows the cutoff for assay positivity. The blue line represents participants with natural SARS-CoV-2 infection based on antibodies against N protein. The red line represents SARS-CoV-2 infection-naive individuals based on the absence of antibodies against N protein. Error bars show SEM. *P* values represent differences between infected and uninfected individuals. *P* values of <0.05 are considered significant.

could be because the saliva samples were taken from the mouth cavity, where IgA is not secreted in abundance (3), and because IgA appears to be an antibody that wanes rapidly (21). Samples from the validation cohort were collected closer to infection onset than samples from the vaccination cohort, which might explain the higher IgA response in the validation cohort (Fig. 3 and Fig. S1). There were detectable IgA levels in the previously infected individuals at 2 months in serum in the vaccination cohort, which was not seen in the infection-naive group.

The fact that we observed only an IgG response in saliva suggests that the antibodies protecting the oral cavity surface come from the systemic response instead of a specific mucosal immune response. It has been shown that immunoglobulins can be transferred from plasma to saliva via the gingival crevice, contributing to the protection of the oral cavity (25, 26). If the antibodies detected in the mucosa come from the blood, it might indicate that a genuine salivary IgA immune response has not been generated.

Vaccinated, previously infected individuals appear to be more resistant to breakthrough infections than vaccinated, infection-naive individuals (18). This could be explained by the previously infected individuals mounting a better mucosal immune response to the virus. Therefore, the issue of breakthrough infections could possibly be circumvented by targeting vaccines against the mucosal surfaces to generate specific IgA mucosal immunity. Consistent with this view are animal studies showing that mucosal boosters seem to mount an efficient mucosal immune response, which provides evidence that it will be possible to achieve sterilizing immunity for at least some time (27–30). However, a robust IgA response does not seem to be generated in saliva. Thus, saliva does not appear to be a good surrogate for mucosal immunity, as suggested elsewhere (17, 31).

Some limitations of our study should be mentioned. We had relatively few individuals who had experienced a natural infection, and we could not control for the timing of such an event. Most of the participants were women. Thus, we cannot exclude the

influence of sex skewing. Unfortunately, due to the design of the experimental setup, it was not possible to test the participants before the collection of the 2-month sample. We did not have a complete follow-up data set at 2 and 6 months after the first vaccination, and we did not sample before and just after vaccination, which might have given an improved picture of the antibody response in saliva. However, a strength of our study is the use of robust assays and uniform saliva sampling and handling, making the results reliable.

Another strength of our study is that it focuses on how the antibody response in saliva wanes from 2 to 6 months after the first vaccine dose, which is different from other studies. This extends the knowledge gap from recent studies, which did not investigate when antibody waning occurs in saliva (22, 32). Thus, our study provides substantial novel information about salivary immunity.

In conclusion, we have shown that IgG is the predominant SARS-CoV-2-specific antibody isotype in saliva 2 months after vaccination, which was elevated in previously infected compared to infection-naive individuals. Levels of IgG in both groups decreased significantly, around 90% between 2 and 6 months after vaccination, suggesting a rapid decline in antibody-mediated saliva immunity against SARS-CoV-2.

## MATERIALS AND METHODS

**Study design, sample collection, and participants.** Four hundred fifty-nine health care personnel from Rigshospitalet and Herlev-Gentofte Copenhagen University Hospitals were invited to participate in this prospective study initiated in December 2020. For power analysis, serum antibody results from previous studies (7, 24) were used due to a lack of studies looking at antibodies in saliva between 2 and 6 months after vaccination. With a power of 80% and an uncertainty level of 5%, we needed at least 65 individuals at each time point in order to get significant results. Demographic data, including age, sex, weight, and height, were collected through an online survey. Participants included in the cohort were all fully vaccinated with two doses of the BNT162b2 COVID-19 vaccine from Pfizer-BioNTech and were part of a longitudinal study reported elsewhere (7, 24). The samples were collected approximately 2 and 6 months after the first vaccine dose. Participants were split into two groups, defined as previously infected and infection-naive based on positivity and negativity for antibodies against N protein, respectively. One participant switched from being N protein seronegative to N protein seropositive between the 2- and the 6-month sampling rounds. Participants receiving vaccines other than BNT162b2 were excluded from the analysis due to the lack of power (20 individuals). One participant was excluded due to missing data regarding sampling dates. Samples from 16 COVID-19-convalescent individuals and 24 infection-naive individuals, who were not included in the main study cohort, were utilized for saliva assay validation. All samples used in the saliva assay validation were collected prior to vaccination. Antibody responses after BNT162b2 vaccination were measured both in saliva and serum. Serum antibody responses were reported previously (7, 24).

**Saliva analyses. (i) Collection of saliva and serum samples.** Saliva samples were collected using Oracol tubes following the manufacturer's instructions (Malvern Medical Developments, Great Britain). Samples were stored at room temperature (RT) for a maximum of 8 h before centrifugation in a Micro-Star centrifuge (VWR, Belgium) at 1,500 × $g$ for 10 min. The supernatant was collected, aliquoted, and stored at −80°C. For every saliva sample, a corresponding venous blood sample was collected at the same time point from the same individual. Blood samples were centrifuged and stored at −80°C.

**(ii) Coupling of beads to the target protein.** Bio-Plex COOH beads (MC10001 to MC10100; Bio-Rad, USA) were mixed for 30 s, followed by 15 s of ultrasonic cleaning in a Branson 200 instrument (Branson Ultrasonics, USA). A total of 100 $\mu$L beads was added to a tube along with the storage solution consisting of phosphate-buffered saline (PBS) with 1% bovine serum albumin (BSA) and 0.05% $NaN_3$. The tube was placed in a magnetic separator (Thermo Fisher Scientific, USA) for 30 s to remove the supernatant. Then, 100 $\mu$L of PBS was added to the tube and vortexed for 30 s, and the tube was placed in a magnetic separator for 30 s. The supernatant was removed, and 80 $\mu$L PBS was added, followed by vortexing for 30 s. Then, beads were incubated with 10 $\mu$L of 50 mg/mL $N$-hydrosulfosuccinimide (S-NHS) and 10 $\mu$L of 50 mg/mL 1-ethyl-3-(3-dimethylaminopropyl) (EDC) (Thermo Scientific, USA) while rotating on an Intelli-Mixer RM-2 rotator (ELMI, Latvia) for 20 min at RT protected from light. After incubation, beads were washed three times with 150 $\mu$L PBS. A total of 6 $\mu$g of SARS-CoV-2 recombinant RBD (produced in HEK293 cells as described elsewhere [33]) was coupled to the beads. Recombinant human serum albumin (rHSA) (AlbIX; kindly provided by Novozymes, Bagsværd, Denmark) (20 $\mu$g) was coupled to the beads as a control for unspecific binding. The protein-bead mixture was incubated for 2 h with rotation. The supernatant was removed, and coupled beads were blocked with PBS–1% BSA for 30 min at RT with rotation, protected from light. Beads were washed with PBS–1% BSA–0.05% $NaN_3$ and stored in this buffer at 4°C, protected from light.

**(iii) Detection of antibodies against SARS-CoV-2 in saliva.** A total of 5,000 coupled beads in PBS–1% BSA (sample buffer) per well were added to Bio-Plex plates (Bio-Rad, USA) and washed twice with sample buffer. A volume of 50 $\mu$L of samples, negative controls, and standards was added and incubated for 1 h at RT. The saliva samples and negative controls were diluted 1:10 in sample buffer

containing 10% skimmed milk. The standards were diluted in a sample buffer, where the IgG standard was diluted 1:500 and the IgA and IgM standards were diluted 1:100. Detection of antibodies was performed by adding 50 $\mu$L of 2 $\mu$g/mL phycoerythrin (PE)-coupled goat anti-human IgG, IgA, and IgM antibodies (204009, 205009, and 202009; all from Bio-Rad, USA) diluted in sample buffer and incubated for 15 min at RT, protected from light.

Wells were filled with 125 $\mu$L sample buffer, briefly shaken on an orbital shaker, and analyzed using the Luminex 200 platform (R&D Systems, USA). The incubation steps were performed at RT on an orbital shaker, protected from light. The plates were washed three times with sample buffer (100 $\mu$L/well) between steps. The final volume during incubations was 50 $\mu$L/well.

The cutoff values were calculated using receiver operating characteristic (ROC) curves, where the specificity was prioritized. The FI cutoffs for positivity in the assays were determined to be 133, 746, and 177 for IgG, IgA, and IgM, respectively.

**Serum analyses. (i) Detection of IgG, IgA, and IgM against RBD in serum.** Serum IgG, IgA, and IgM antibodies against RBD were measured using a nationally validated in-house ELISA-based assay, as previously described (33). In short, Nunc-Maxisorp 384-well plates (Thermo Fisher Scientific) were coated with recombinant RBD (rRBD) in PBS and incubated overnight (ON) at 4°C. Plates were blocked with PBS–0.05% Tween (Merck) (PBS-T) for 1 h. Diluted serum samples were added in PBS-T–5 mM EDTA–5% skimmed milk and incubated for 1 h. Antibodies were detected using horseradish peroxidase (HRP)-conjugated polyclonal rabbit-anti-human IgG, IgA, and IgM (P0214, P2016, and P0215, respectively; all from Agilent Technologies) for 1 h at RT. Plates were developed for 7 (IgG) or 10 (IgM and IgA) minutes with TMB-One (Kem-En-Tec) as the substrate, and the reaction was stopped with a solution of 0.3 M $H_2SO_4$. A Synergy HT absorbance reader (Biotek Instruments) was used to measure the optical density (OD) at 450 to 630 nm. A mix of recombinant human IgG, IgA, and IgM against SARS-CoV-2 RBD (1:2,000 dilution; A02038-100, A02071-100, and A02046-100; all from GenScript) was used as a calibrator. The plates were washed four times with PBS-T between steps. The incubations were all done at RT on an orbital shaker. The cutoffs for positivity in the assay were 225, 100, and 200 AU/mL for IgG, IgA, and IgM, respectively.

IgG, IgA, and IgM levels were interpolated using nonlinear regression with four-parameter curve fitting, and antibody levels were reported in $\log_{10}$(FI) and $\log_{10}$(AU/mL) for saliva and serum, respectively.

**(ii) Detection of antibodies against nucleocapsid protein in serum.** The Elecsys anti-SARS-CoV-2 assay (Roche Diagnostics) was employed to test for previous infection with SARS-CoV-2. This electrochemiluminescence immunoassay (ECLIA) detects antibodies against N protein, which is not part of the BNT162b2 vaccine. The analyses were carried out on the Cobas 8000 platform (e801 module) following the manufacturer's instructions.

**Statistical analyses.** Statistical analyses were performed using R (version 4.1.0 for Windows; R Foundation for Statistical Computing) and GraphPad Prism (version 9.3.1). In the saliva assay validation, values were $\log_{10}$ transformed, and the differences in antibody levels between samples from COVID-19-convalescent and infection-naive unvaccinated persons were assessed using the Mann-Whitney U test. To summarize the characteristics of the vaccination cohort, continuous data were reported as medians with IQRs or as means with SEM, as appropriate. Categorical data were reported as frequency counts and percentages of individuals within each category. Student's $t$ test was used to calculate the statistical difference between the previously infected and infection-naive groups at one time point and between the two groups at different time points. The Spearman rank rho test was used to assess antibody correlations between saliva and serum.

**Data availability.** Data will be made available upon request.

## SUPPLEMENTAL MATERIAL

Supplemental material is available online only.

**SUPPLEMENTAL FILE 1**, PDF file, 0.04 MB.

## ACKNOWLEDGMENTS

This work was financially supported by grants from the Carlsberg Foundation (CF20-476 0045, granted to P.G.), the Novo Nordisk Foundation (NFF205A0063505 and NNF20SA0064201, granted to P.G.), and the Svend Andersen Research Foundation (SARF2021, granted to P.G.).

We thank Mads Engelhardt Knudsen and Sif Kaas Nielsen from the Laboratory of Molecular Medicine at Rigshospitalet for their excellent technical assistance in processing and analyzing the samples. Lisbeth Andreasen, Annie Mørk, Ann Kristine Thorsteinsson, and Tung Thanh Phan from the Department of Clinical Biochemistry at Rigshospitalet are thanked for their excellent technical assistance in analyzing the samples for antibodies against N protein. We also thank Alexandra Rosengaard Röthlin Eriksen from the Department of Emergency Medicine, Herlev and Gentofte Hospital, for her logistics and sample collection assistance. The Danish COVID-19 Biobank (part of Bio- and Genome Bank Denmark) is acknowledged for serum samples and for data regarding handling and storage.

S.D.N., K.K.I., H.B., and P.G. conceived and designed the study. B.E.H., L.P.-A., C.B.H., R.F.-S., and L.H. performed experiments. J.R.M., L.P.-A., and P.G. analyzed the data. K.F., M.M.P.-H., D.L.M., L.D.H., R.B.H., S.R.H., S.D.N., K.K.I., and H.B. collected samples and clinical data. B.E.H., R.B.-O., and A.R. produced recombinant proteins and enabled assay development. J.R.M. and P.G. wrote the first draft of manuscript with subsequent input from the coauthors. All coauthors approved the final version of the manuscript.

We declare no competing interests.

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
