## [Reviewer comments · Microbiology Spectrum]

Microbiology Spectrum

Short-lived antibody-mediated saliva immunity against SARS-CoV-2 after vaccination

Johannes Madsen, Bettina Holm, Laura Pérez-Alós, Rafael Bayarri-Olmos, Anne Rosbjerg, Kamille Fogh, Mia Pries-Heje, Dina Møller, Cecilie Hansen, Line Heftdal, Rasmus Hasselbalch, Sebastian Hamm, Ruth Frikke-Schmidt, Linda Hilsted, Susanne Nielsen, Kasper Iversen, Henning Bundgaard, and Peter Garred

Corresponding Author(s): Peter Garred, Rigshospitalet

Review Timeline:

Submission Date:	December 1, 2022
Editorial Decision:	January 30, 2023
Revision Received:	February 3, 2023
Accepted:	February 9, 2023

Editor: Takamasa Ueno

Reviewer(s): The reviewers have opted to remain anonymous.

Transaction Report:

DOI: <https://doi.org/10.1128/spectrum.04947-22>

January 30, 2023

Prof. Peter Garred
University of Copenhagen
Department of Clinical Immunology
Copenhagen
Denmark

Re: Spectrum04947-22 (Short-lived antibody-mediated saliva immunity against SARS-CoV-2 after vaccination)

Dear Prof. Peter Garred:

Our reviewers suggested modifications of the current form of your manuscript based on the points shown below. In particular, both ones strongly suggested to improve writing and correct grammatical errors throughout the manuscript when you resubmit the revised version.

Link Not Available

Sincerely,

Takamasa Ueno

Journals Department
Reviewer comments:

Reviewer #1 (Comments for the Author):

This study examines the antibody response in saliva and serum 2 and 6 months after the first BNT162b2 vaccine. The study found that vaccinated infected individuals had higher antibody levels in saliva at 2 months, but levels declined for both groups after 6 months. The study also found that IgA and IgM antibodies were hardly detectable in saliva at any time points. The study concluded that the salivary antibody response was short-lived and declined to almost undetectable levels after 6 months.

Comments.

1. To avoid confusion, the use of the term "infected individuals/group" to refer to vaccinated individuals with positive antibodies

against the N-protein should be rephrased to avoid any confusion with individuals with natural immunity.

2. The article would benefit from the inclusion of demographic data and characteristics of the individuals in the "vaccinated infected" and "vaccinated uninfected" groups, as this would aid in interpreting the results of the study.
3. The conclusion statement regarding the duration of salivary immunity against SARS-CoV-2 after both natural infection and systemic vaccination should be revised as the data presented in the study does not fully support the extrapolation made.
4. The manuscript contains several grammatical errors, including incorrect use of terms. Additionally, the language and flow of the methodology sections may be improved for better readability.

Reviewer #2 (Comments for the Author):

Johannes Roth Madsen and colleagues presented data which show a short-lived antibody-mediated saliva immunity against SARS-CoV-2 after vaccination.

The manuscript is written in a comprehensive way and I have only minor, but important, comments, especially #5 and #8:

1. page (p) 8: saliva analyses (as these are several)
2. p8: reasons for 89% females in the study should be given. Healthcare personnel is nowadays dominated by females, but not really 9:1.
3. p13: the headlines under "Serum analysis" - which should read "Serum analyses" - end with a ":", which is an unusual format and thus likely not journal style.
4. p13: "people" is an unusual way to describe "participants" or "subjects".
5. p13, last three lines - proof-reading by the co-authors was somewhat sloppy, because it obviously must read "... (Figure 2c and 2e, respectively). After 6 months, almost the same correlation was observed in the infected group ($\rho=0.53$, $p=0.052$), but was lost in the uninfected group ($\rho=0.14$, $p=0.055$) (Figure 2d and 2f, respectively)."
6. p14: The study investigates IgG, IgA and IgM in saliva and serum. It is therefore strange, why the IgM data are in a supplementary figure!?
7. p14: The text is partially used to describe figures "Figure 3a depicts...., Figure 3b depicts" which is, unfortunately, generally getting more and more common, but of course wrong: the legends should describe a figure and a statement in the text should be followed by the figure number in parenthesis.
As an example: "The infected group had a mean serum IgA level against RBD of 2.82 compared to 1.74 ... in the uninfected group (p....; **Fig. 3b**).
8. p27 & p29. Again, sloppy proof-reading for both legends: not the antibodies against protein N determine the uninfected subjects (just the contrary) and it is the absence of antibodies against protein N which excludes a previous contact with the virus: "The red line represents SARS-CoV-2 infectious naïve **subjects** based on **the absence of** antibodies against protein N."

Staff Comments:

Preparing Revision Guidelines

Please return the manuscript within 60 days; if you cannot complete the modification within this time period, please contact me. If you do not wish to modify the manuscript and prefer to submit it to another journal, please notify me of your decision immediately so that the manuscript may be formally withdrawn from consideration by Microbiology Spectrum.

If your manuscript is accepted for publication, you will be contacted separately about payment when the proofs are issued; please follow the instructions in that e-mail. Arrangements for payment must be made before your article is published. For a

complete list of **Publication Fees**, including supplemental material costs, please visit our website.

Response to Reviewers

We appreciate the effort made by the reviewers to improve this manuscript. Here, we address one by one the questions and comments raised by Reviewer #1 and Reviewer #2.

Reviewer comments

Reviewer #1 (Comments for the Author):

This study examines the antibody response in saliva and serum 2 and 6 months after the first BNT162b2 vaccine. The study found that vaccinated infected individuals had higher antibody levels in saliva at 2 months, but levels declined for both groups after 6 months. The study also found that IgA and IgM antibodies were hardly detectable in saliva at any time points. The study concluded that the salivary antibody response was short-lived and declined to almost undetectable levels after 6 months.

Comments.

Reviewer 1 comment #1. To avoid confusion, the use of the term "infected individuals/group" to refer to vaccinated individuals with positive antibodies against the N-protein should be rephrased to avoid any confusion with individuals with natural immunity.

Authors reply #1. Thank you for the comment. We have updated the manuscript accordingly and we are now referring to the "infected individuals" as individuals with hybrid immunity (vaccination + previous infection) and "uninfected individuals" as vaccinated infection-naïve individuals unless vaccination is clearly stated in the sentence.

In the section regarding saliva assay validation, all samples were collected prior to vaccination, and we are referring to the groups as unvaccinated COVID-19 convalescent individuals and infection-naïve individuals.

Moreover, we have updated titles and legends in Figure 2 and Supplementary Figure 1 accordingly to avoid confusions.

Reviewer 1 comment #2. The article would benefit from the inclusion of demographic data and characteristics of the individuals in the "vaccinated infected" and "vaccinated uninfected" groups, as this would aid in interpreting the results of the study.

Authors reply #2. We acknowledge the benefit of the inclusion of demographic data and individual characteristics of each vaccinated group (vaccinated, previously infected; and vaccinated, infection-naïve). Therefore, we have modified accordingly Table 1.

Reviewer 1 comment #3. The conclusion statement regarding the duration of salivary immunity against SARS-CoV-2 after both natural infection and systemic vaccination should be revised as the data presented in the study does not fully support the extrapolation made.

Authors reply #3. We apologize for the ambiguous phrasing in the conclusion statement. We have therefore rephrased it as follows in the discussion:

“In conclusion, we have shown that IgG is the predominant SARS-CoV-2 specific antibody isotype in saliva 2 months after vaccination, which was elevated in previously infected compared to infection-naïve individuals. Levels of IgG in both groups decreased significantly with around 90% between 2 and 6 months after vaccination, suggesting a rapid decline in antibody-mediated saliva immunity against SARS-CoV-2.”.

Reviewer 1 comment #4. The manuscript contains several grammatical errors, including incorrect use of terms. Additionally, the language and flow of the methodology sections may be improved for better readability.

Authors reply #4. We understand the reviewer’s concern regarding the language quality of the manuscript. We have rewritten most part of it to correct the grammar and to improve the reading flow.

Reviewer #2 (Comments for the Author):

Johannes Roth Madsen and colleagues presented data which show a short-lived antibody-mediated saliva immunity against SARS-CoV-2 after vaccination.

The manuscript is written in a comprehensive way and I have only minor, but important, comments, especially #5 and #8:

Reviewer 2 comment #1. page (p) 8: saliva analyses (as these are several)

Authors reply #1. Corrected.

Reviewer 2 comment #2. p8: reasons for 89% females in the study should be given. Healthcare personnel is nowadays dominated by females, but not really 9:1.

Authors reply #2. We are aware of the high percentage of females in our cohort, which we address as a limitation in the discussion section. It is true that the average percentage of females in healthcare personnel is approximately 76-80% in the US and Europe. However, as this study was opened to all healthcare personnel to volunteer to participate, we cannot discard higher participation from personnel from specific job positions, such as nurses or midwives, where the presence of females is even higher. Nevertheless, we are unknown of the specific job position of the participants, but this might be the most likely reason for the higher proportion of females in this study.

Reviewer 2 comment #3. p13: the headlines under "Serum analysis" - which should read "Serum analyses" - end with a ":", which is an unusual format and thus likely not journal style.

Authors reply #3. Corrected.

Reviewer 2 comment #4. p13: "people" is an unusual way to describe "participants" or "subjects".

Authors reply #4. Corrected.

Reviewer 2 comment #5. p13, last three lines - proof-reading by the co-authors was somewhat sloppy, because it obviously must read "... (Figure 2c and 2e, respectively). After 6 months, almost the same correlation was observed in the infected group ($\rho=0.53$, $p=0.052$), but was lost in the uninfected group ($\rho=0.14$, $p=0.055$) (Figure 2d and 2f, respectively).)

Authors reply #5. We apologize for this obvious mistake. We have corrected the labeling accordingly.

Reviewer 2 comment #6. p14: The study investigates IgG, IgA and IgM in saliva and serum. It is therefore strange, why the IgM data are in a supplementary figure!?

Authors reply #6. We understand the reviewer's concern regarding the IgM results. We have moved Supplementary Figure 2 from the supplement into the main text (newly name Figure 4).

Reviewer 2 comment #7. p14: The text is partially used to describe figures "Figure 3a depicts...., Figure 3b depicts" which is, unfortunately, generally getting more and more common, but of course wrong: the legends should describe a figure and a statement in the text should be followed by the figure number in parenthesis. As an example: "The infected group had a mean serum IgA level against RBD of 2.82 compared to 1.74 ... in the uninfected group (p.....; Fig. 3b).

Authors reply #7. We appreciate the reviewer's suggestion and we have changed the figure's description accordingly.

Reviewer 2 comment #8. p27 & p29. Again, sloppy proof-reading for both legends: not the antibodies against protein N determine the uninfected subjects (just the contrary) and it is the absence of antibodies against protein N which excludes a previous contact with the virus: "The red line represents SARS-CoV-2 infectious naïve subjects based on the absence of antibodies against protein N."

Authors reply #8. Corrected.

February 9, 2023

Prof. Peter Garred
Rigshospitalet
Department of Clinical Immunology
Copenhagen
Denmark

Re: Spectrum04947-22R1 (Short-lived antibody-mediated saliva immunity against SARS-CoV-2 after vaccination)

Dear Prof. Peter Garred:

Your manuscript has been accepted, and I am forwarding it to the ASM Journals Department for publication. You will be notified when your proofs are ready to be viewed.

Sincerely,

Takamasa Ueno
Editor, Microbiology Spectrum
